

# One-dimensional Fermi polaron after a kick:
# Two-sided singularity of the momentum distribution,
# Bragg reflection and other exact results

**Oleksandr Gamayun⋆ and Oleg Lychkovskiy**

London Institute for Mathematical Sciences, Royal Institution,
21 Albemarle St, London W1S 4BS, UK

⋆ og@lims.ac.uk

## Abstract

A mobile impurity particle immersed in a quantum fluid forms a polaron – a quasiparticle consisting of the impurity and a local disturbance of the fluid around it. We ask what happens to a one-dimensional polaron after a kick, i.e. an abrupt application of a force that instantly delivers a finite impulse to the impurity. In the framework of an integrable model describing an impurity in a one-dimensional gas of fermions or hard-core bosons, we calculate the distribution of the polaron momentum established when the post-kick relaxation is over. A remarkable feature of this distribution is a two-sided power-law singularity. It emerges due to one of two processes. In the first process, the whole impulse is transferred to the polaron, without creating phonon-like excitations of the fluid. In the second process, the impulse is shared between the polaron and the center-of-mass motion of the fluid, again without creating any fluid excitations. The latter process is, in fact, a Bragg reflection at the edge of the emergent Brillouin zone. We carefully analyze the conditions for each of the two processes. The asymptotic form of the distribution in the vicinity of the singularity is derived.

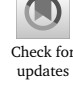

# 1 Introduction

The behavior of an impurity particle propagating in a host media is a paradigmatic problem in physics. To address this problem, a concept of polaron was introduced by Landau [1] and Pekar [2] at the down of the quantum theory of condensed matter. A polaron is a quasiparticle consisting of the impurity along with the local disturbance of the host media cased by the interaction between the impurity and host particles. Polaron properties, such as mass or dispersion relation, can be quite different from that of the bare impurity [3]. The polaron framework can be universally applied to virtually any combination of impurity particle and host media [4–7].

Current experimental advance in the ultracold atomic gases allows one to create and control two-component mixtures with large concentration imbalance, thus offering a novel, extremely flexible platform for studying physics of polarons [8–15]. Importantly, such experiments can be performed in the reduced spatial geometries, where effects of interactions are more pronounced.

One-dimensional polarons are particularly remarkable. Their distinctive feature is the ability to move perpetually at zero temperature – an effect reminiscent but not identical to superfluidity [16–18]. This effect is universal and stems from the non-trivial spectral edge of any one-dimensional fluid [19]. It implies that the polaron momentum is a bona-fide quantum number (at least at zero temperature).

A steadily increasing deal of attention is being attracted by non-equilibrium aspects of the polaron formation and dynamics [20–46]. Valuable insights into the nonequilibrium polaron physics [16, 22, 33] come from studies of a one-dimensional integrable model introduced in 1965 by McGuire [47, 48]. Importantly, integrability facilitates non-perturbative analytical investigation of the strong correlation effects hardly available otherwise.

In this paper, we consider the effect of kicking the polaron in the McGuire model at zero temperature. In the context of cold atom experiments, the kick can be realized by photon scattering or absorption, or by moving optical tweezers [49]. On a formal level, the kick corresponds to applying an external force to the impurity within a short time interval, so that the impurity acquires a finite impulse. The kick is followed by a relaxation, when the acquired momentum is shared between the polaron and the excitations of the fluid created by the kick. The relaxation is effectively over when all created fluid excitations have detached and moved

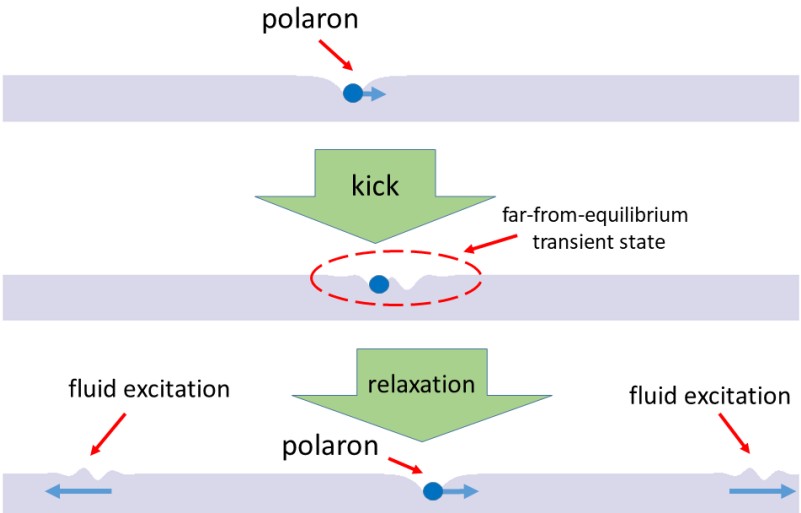

Figure 1: A cartoon of the out-of-equilibrium protocol under study. A polaron, initially in equilibrium, is kicked by an instant application of a force to the impurity. The kick changes the polaron momentum and creates excitations of the fluid. We describe the steady state of the polaron after the relaxation is over and all fluid excitations have broken apart.

away from the polaron, see Fig. 1. We calculate and analyze the probability distribution of the polaron momentum in the thus established (quasi-)steady state.

The rest of the paper is structured as follows. In the next section we provide a concise but self-contained description of the McGuire model and its Bethe Ansatz solution, with the focus on the polaron properties. This section is based on the prior literature [47, 50–52]. Original results are presented in Section 3. There we provide the polaron momentum distribution and analyze its singularity structure. The last section discusses the implications of the results in a broader context. Derivations and proofs can be found in the Appendix.

## 2 McGuire model and its Bethe Ansatz solution

### 2.1 The model

We use a simple and yet non-trivial integrable model of a polaron. This model was introduced by McGuire who obtained its Bethe-Ansatz solution in 1965 [47, 48]. Later the McGuire model turned out to represent a specific sector of a more general Yang-Gaudin model [53, 54]. Much later it was realized that the Bethe eigenstates of the McGuire model can be represented as Slater-like determinants [16, 50, 55]. This discovery opened an avenue for a flurry of exact results [16, 22, 33, 50–52, 56–59]. In the present section we review the McGuire model and its solution. The exposition mostly follows ref. [51].

McGuire model describes a single impurity particle immersed into a one-dimensional gas of $N$ spinless fermions (or, equivalently [19], hard-core bosons), the mass of the impurity and a fermion being the same. Fermions do not interact one with another but interact with the

impurity. We work in the first quantization, where the Hamiltonian of the model reads

$$H = \frac{1}{2}P_{\text{imp}}^2 + \frac{1}{2}\sum_{j=1}^{N}P_j^2 + \frac{2p_{\text{F}}}{\alpha}\sum_{j=1}^{N}\delta(x_j - x_{\text{imp}}).$$ (1)

Here, $x_j$ ($P_j$) is the coordinate (momentum) of the $j$'th fermion, $j = 1, \dots, N$, and $x_{\text{imp}}$ ($P_{\text{imp}}$) is that of the impurity. Translation invariance is imposed by introducing periodic boundary conditions with the circumference $L$. Any wave function should be periodic in any coordinate with the period $L$ and antisymmetric in fermionic coordinates. The number of fermions, $N$, is assumed to be odd. The Fermi momentum is defined as $p_{\text{F}} = \pi(N-1)/L$. We will be interested in the thermodynamic limit of $N, L \to \infty$ with $p_{\text{F}}$ being fixed. Only the case of repulsion will be considered here, which corresponds to a positive interaction strength $(2pF)/\alpha$.

The total momentum $P_{\text{tot}} = P_{\text{imp}} + \sum_{j=1}^{N}P_j$ is an integral of motion of the model. Its eigenvalues (denoted by the same symbol $P_{\text{tot}}$) are quantized with the momentum quantum $2\pi/L$,

$$P_{\text{tot}} = \frac{2\pi}{L}M,$$ (2)

where $M$ is an integer.

## 2.2 Bethe Ansatz

The eigenstates of the McGuire model are labelled by $N + 2$ integers. One of them is $M$, and others are organized in an ordered set $\mathbf{n} = \{n_1, n_2, \dots n_{N+1}\}$, $n_1 < n_2 < \cdots < n_{N+1}$ satisfying the constraint

$$\sum_{l=1}^{N+1}n_l \in [M, M+N].$$ (3)

An eigenstate $|\mathbf{n}, M\rangle$ is given by

$$|\mathbf{n}, M\rangle = \mathcal{N}\, e^{iP_{\text{tot}}x_{\text{imp}}}
\begin{vmatrix}
e^{ik_1 y_1} & e^{ik_2 y_1} & \dots & e^{ik_{N+1}y_1} \\
e^{ik_1 y_2} & e^{ik_2 y_2} & \dots & e^{ik_{N+1}y_2} \\
\cdot & \cdot & \cdot & \cdot \\
\cdot & \cdot & \cdot & \cdot \\
e^{ik_1 y_N} & e^{ik_2 y_N} & \dots & e^{ik_{N+1}y_N} \\
e^{-i\delta_1}\sin\delta_1 & e^{-i\delta_2}\sin\delta_2 & \dots & e^{-i\delta_{N+1}}\sin\delta_{N+1}
\end{vmatrix},$$ (4)

where $\mathcal{N}$ is a normalization constant, $y_j \equiv (x_j - x_{\text{imp}})\bmod L$ is the position of the $j$'th fermion relative to the position of the impurity, and $(N + 1)$ pseudomomenta $k_l$ are fixed by integers $n_l$ up to phase shifts $\delta_l \in [0, \pi)$,

$$k_l = \frac{2\pi}{L}\left(n_l - \frac{\delta_l}{\pi}\right), \qquad l = 1, 2, \dots, N+1.$$ (5)

The phase shifts $\delta_l$ should be found from Bethe equations. We do not need the complete set of Bethe equations since in the thermodynamic limit and for eigenstates in the *bottom of the spectrum* (whose precise meaning is discussed in the next subsection) the it reduces to a single equation on an auxiliary but very important variable – *polaron rapidity* $\Lambda \in (-\infty, +\infty)$. This is the equation (8) introduced in the next subsection. It has a single root $\Lambda_{\mathbf{n},M}$ whenever the constraint (3) is satisfied, and no roots otherwise. The $(N + 1)$ phase shifts are expressed through this root as

$$\delta_l = \frac{\pi}{2} - \arctan\left(\Lambda_{\mathbf{n},M} - \frac{2\pi}{L}\alpha\, n_l\right),$$ (6)

up to corrections negligible in the thermodynamic limit.

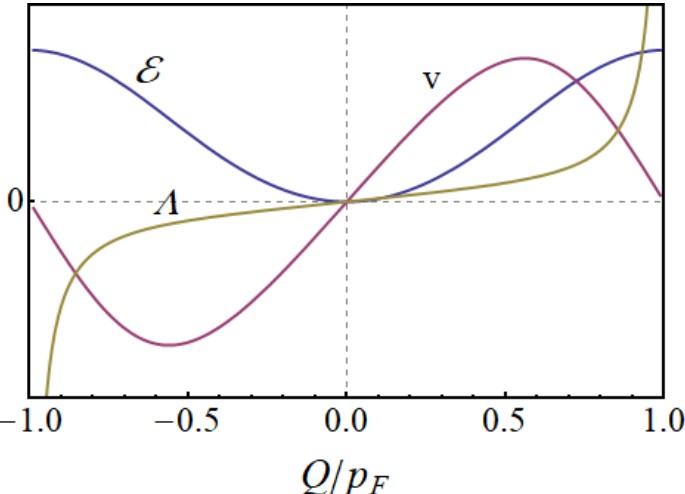

Figure 2: Schematic dependence of the polaron energy $\mathcal{E}$, velocity $v$ and rapidity $\Lambda$ on the polaron momentum $Q$. The units of the vertical axis are arbitrary. $\mathcal{E}(Q)$ is shifted by a constant to ensure $\mathcal{E}(0) = 0$.

## 2.3 Spectrum and a Bethe equation in the thermodynamic limit

The total momentum $P_{\text{tot}}$ and energy $E_{\text{tot}}$ of the eigenstate $|\mathbf{n}, M\rangle$ can be expressed through the corresponding pseudomomenta,

$$P_{\text{tot}} = \sum_{l=1}^{N+1} k_l, \qquad E_{\text{tot}} = \frac{1}{2} \sum_{l=1}^{N+1} k_l^2. \tag{7}$$

In fact, the right hand sides (r.h.s.) of these equations give the expectation values of operators $P_{\text{tot}}$ and $H$, respectively, for a state of the form (4) for arbitrary $k_l$, not necessarily the solutions of Bethe equations. In the thermodynamic limit, one can obtain the sole relevant Bethe equation on the rapidity by plugging expressions (5),(6) into the first equation (7) and replacing the sum by the integral. The resulting equation on $\Lambda$ reads

$$Q(\Lambda) = \frac{2\pi}{L} \left( M - \sum_{l=1}^{N+1} \left( n_l - \frac{1}{2} \right) \right), \tag{8}$$

where the function $Q(\Lambda)$ is defined as

$$\frac{Q(\Lambda)}{p_{\text{F}}} = \frac{1}{\pi\alpha} \left( (\Lambda + \alpha) \arctan(\Lambda + \alpha) - (\Lambda - \alpha) \arctan(\Lambda - \alpha) + \frac{1}{2} \log \frac{1 + (\alpha - \Lambda)^2}{1 + (\alpha + \Lambda)^2} \right). \tag{9}$$

As we discuss in the next subsection, $Q(\Lambda)$ is interpreted as the *polaron momentum*. It is a monotonically increasing function of $\Lambda$ (see Fig. 2), therefore the equation (8) has at most one root $\Lambda_{\mathbf{n},M}$. Further, $Q(\Lambda) \in [-p_{\text{F}}, p_{\text{F}}]$, which implies that a root exists whenever the constraint (3) is satisfied.

For a given total momentum, one can define a (momentum-dependent) ground state. Its energy is denoted by $\mathcal{E}(P_{\text{tot}})$. The latter function constitutes the lower edge of the many-body spectrum. In the thermodynamic limit, it is periodic with the period $2p_{\text{F}}$. The spectrum is therefore divided into Brillouin zones of width $2p_{\text{F}}$. The main Brillouin zone corresponds to $P_{\text{tot}} \in [-p_{\text{F}}, p_{\text{F}}]$.

The set $\mathbf{n}$ of each ground state consists of consecutive integers. All eigenstates within a Brillouin zone share the same set $\mathbf{n}$ but have different $M$. The main Brillouin zone corresponds to the set $\mathbf{n} = \{-(N-1)/2, \ldots, (N+1)/2\}$. This implies $P_{\text{tot}} = Q(\Lambda_{\mathbf{n},M})$ for each

ground state $|\mathbf{n}, M\rangle$ from the main Brillouin zone. Other Brillouin zones have total momentum $P_{\text{tot}} = Q(\Lambda_{\mathbf{n},M}) + 2m\,p_{\text{F}}$, where an integer $m$ is the number of the zone. The momentum shift $(2m\,p_{\text{F}})$ corresponds to the center-of-mass motion of the Fermi gas.

We say that an eigenstate is in the *bottom of the spectrum* whenever its energy differs from $\mathcal{E}(P_{\text{tot}})$ by a value that is $O(1)$ in the thermodynamic limit. The set $\mathbf{n}$ for such an eigenstate differs from that for a ground state by "particle-hole excitations" whose number is $o(N)$ in the thermodynamic limit.

## 2.4 Fermi polaron

The impurity along with the disturbance of the Fermi gas around it is commonly known as Fermi polaron (another term "depleton" is also used in the one-dimensional context [60–62]). In the McGuire model the polaron can be characterized in a remarkably precise and rigorous way. Namely, it turns out that, for any eigenstate in the bottom of the spectrum, any local property of the impurity (e.g. its momentum distribution, static correlation function *etc*) depends only on the corresponding polaron rapidity. In other words, if the rapidities of two eigenstates $|\mathbf{n}, M\rangle$ and $|\mathbf{n}', M'\rangle$ are identical up to finite size corrections, $\Lambda_{\mathbf{n}',M'} = \Lambda_{\mathbf{n},M} + O(1/N)$, then local properties of the impurity in these two states are also identical up to finite size corrections. Therefore one can consistently interpret an eigenstate $|\mathbf{n}, M\rangle$ as containing a polaron with the rapidity $\Lambda_{\mathbf{n},M}$ and a certain number of Fermi sea excitations. The latter do not alter local polaron properties since their density vanishes in the thermodynamic limit.

Ground states is the main Brillouin zone contain only a polaron, without additional excitations of the Fermi sea. For this reason the momentum $Q(\Lambda)$ and the energy $\mathcal{E}(Q(\Lambda)) = \mathcal{E}(\Lambda)$ of such ground states are interpreted as the polaron momentum and energy, respectively. The polaron energy is explicitly given by

$$\frac{\mathcal{E}(\Lambda)}{p_{\text{F}}^2} = \frac{1}{\pi\alpha} - \frac{1+\alpha^2-\Lambda^2}{2\pi\alpha^2}\Big(\arctan(\Lambda+\alpha) + \arctan(\Lambda-\alpha)\Big) + \frac{\Lambda}{2\pi\alpha^2}\log\frac{1+(\alpha-\Lambda)^2}{1+(\alpha+\Lambda)^2}. \quad (10)$$

A distinct feature of a polaron in one dimension is that it can move perpetually with a velocity below a critical one (no matter whether the model is integrable or not) [16–18, 22]. The critical velocity does not exceed the speed of sound in the medium hosting the polaron [17, 18]. The velocity operator is defined through the Heisenberg equation as $i[H, x_{\text{imp}}]$ and, for the Hamiltonian (1), coincides with the impurity's momentum operator $P_{\text{imp}}$.

In the McGuire model the impurity's velocity $v(\Lambda)$ is expressed through the polaron rapidity as

$$\frac{v(\Lambda)}{v_{\text{F}}} = \langle \mathbf{n}, M|P_{\text{imp}}|\mathbf{n}, M\rangle = \frac{\Lambda}{\alpha} + \frac{1}{2\alpha}\frac{\log\frac{1+(\alpha-\Lambda)^2}{1+(\alpha+\Lambda)^2}}{\arctan(\alpha-\Lambda) + \arctan(\alpha+\Lambda)}, \quad (11)$$

where $v_{\text{F}} = p_{\text{F}}$ is the Fermi velocity, see Fig. 2. One can verify that the polaron velocity satisfies the usual relation for the group velocity,

$$v(\Lambda) = \frac{\partial\mathcal{E}}{\partial\Lambda}\left(\frac{\partial Q}{\partial\Lambda}\right)^{-1} = \frac{\partial\mathcal{E}}{\partial Q}. \quad (12)$$

Explicit formula for the complete velocity distribution of an impurity in a polaron eigenstate can be found in ref. [52].

Since $Q(\Lambda)$ is a monotonic function, it can be inverted, $\Lambda = \Lambda(Q)$. As a consequence, the polaron can be unambiguously labelled not only by its rapidity $\Lambda \in (-\infty, \infty)$ but also by its momentum $Q \in [-p_{\text{F}}, p_{\text{F}}]$.

# 3 Polaron after a kick

## 3.1 Distribution over polaron rapidities after a general quantum quench

A quantum quench initializes a system in an out-of-equilibrium state $|\text{in}\rangle$. The quench is followed by relaxation and, eventually, by establishing a post-quench equilibrium state. The long-time expectation value $\mathcal{O}_\infty$ of an observable $\mathcal{O}$ can be obtained by averaging the observable over the *diagonal ensemble* [63]. Specifically, employing our notations for eigenstates, one obtains

$$\mathcal{O}_\infty = \sum_{|\mathbf{n},M\rangle} \left| \langle \mathbf{n}, M | \text{in} \rangle \right|^2 \langle \mathbf{n}, M | \mathcal{O} | \mathbf{n}, M \rangle. \tag{13}$$

On physical grounds, one expects that a local quench excites only eigenstates in the bottom of the spectrum (numerical verification of this statement for a different local quench in the same model can be found in refs. [33,64]). As discussed above, for such states the diagonal matrix element of any polaron observable depends only on the polaron momentum, $\langle \mathbf{n}, M | \mathcal{O} | \mathbf{n}, M \rangle = \mathcal{O}(\Lambda_{\mathbf{n},M})$. Therefore, it makes sense to rewrite eq. (13) as

$$\mathcal{O}_\infty = \int_{-\infty}^{\infty} dQ \, \Gamma(\Lambda) \, \mathcal{O}(\Lambda), \tag{14}$$

where

$$\Gamma(\Lambda) = \sum_{|\mathbf{n},M\rangle} \left| \langle \mathbf{n}, M | \text{in} \rangle \right|^2 \delta(\Lambda - \Lambda_{\mathbf{n},M}), \tag{15}$$

is the probability distribution of the equilibrium post-quench state over polaron rapidities.

Eq. (14) allows for a transparent physical interpretation. The post-quench equilibrium state features a polaron of rapidity $\Lambda$ with the probability $\Gamma(\Lambda)$. In addition, this state contains Fermi sea excitation that, however, break off far apart from the polaron (since their velocity always exceeds that of the polaron [17,18,65]) and thus have no effect on the observables related to the impurity.

## 3.2 Preparing the initial state by kicking the impurity

In the present paper we consider a specific way to prepare the initial out-of-equilibrium state. It consists of applying a large force $F$ to the impurity over a small time interval $\tau$. We consider the limit of $F \to \infty$, $\tau \to 0$ with the delivered impulse $\Delta P = F\tau$ fixed. This may be viewed as an instant kick applied to the impurity.

We assume that prior to the kick the system is in an eigenstate $|\mathbf{n}^0, M^0\rangle$ with the polaron rapidity $\Lambda_0 \equiv \Lambda_{\mathbf{n}^0, M^0}$ and momentum $Q_0 = Q(\Lambda_0)$. This eigenstate is assumed to belong to the bottom of the spectrum. Naively, kicking the impurity at time $t = 0$ can be described by adding the term $-F x_{\text{imp}} \delta(t/\tau)$ to the Hamiltonian (1). Then the out-of-equilibrium state immediately after the kick reads

$$|\text{in}\rangle = e^{i \Delta P x_{\text{imp}}} |\mathbf{n}^0, M^0\rangle. \tag{16}$$

The above simple consideration is not rigorous since the linear potential breaks the translation invariance and is incompatible with periodic boundary conditions. Nevertheless, eq. (16) remains correct, provided $\Delta P$ is an integer of momentum quanta $2\pi/L$ which we assume in what follows. We justify eq. (16) in Appendix A by employing a more elaborate (although somewhat cumbersome) argument.

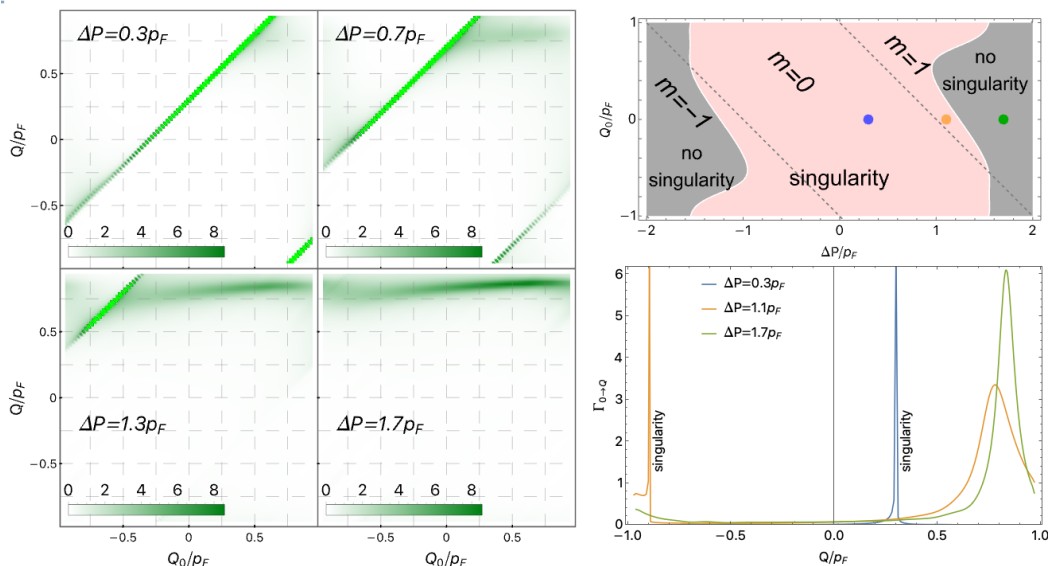

Figure 3: **Left:** Color plots of $\Gamma_{Q_0 \to Q}$ for various impulses $\Delta P$. Bragg reflection is clearly seen for smaller values of $\Delta P$. The bright green color indicates the position of the singularity. **Top right:** The range of the initial polaron momentum $Q_0$ and impulse $\Delta P$ where the post-kick polaron momentum distribution $\Gamma_{Q_0 \to Q}$ has a two-sided power law singularity. Points indicate values of $Q_0$ and $\Delta P$ chosen for the bottom right plot. Dashed lines mark the boundaries of the Brillouin zone. **Bottom right:** Three typical shapes of $\Gamma_{Q_0 \to Q}$ as a function of $Q$ (with $Q_0 = 0$ and values $\Delta P$ specified in the plot). For $\Delta P = 0.3 p_F$ the kick acts within the main Brillouin zone and $\Gamma_{Q_0 \to Q}$ features a sharp singularity at $Q = Q_0 + \Delta P$. For $\Delta P = 1.1 p_F$ the Bragg reflection from the Brillouin zone boundary occurs, the singularity at $Q = Q_0 + \Delta P - 2 p_F$ is accompanied by a broad maximum on the opposite side of the Brillouin zone. For $\Delta P = 1.7 p_F$ the singularity is absent. The coupling constant is $\alpha = 2$.

### 3.3 Polaron momentum distribution after the kick

Here we present the result for the polaron rapidity distribution established as a result of relaxation after the kick. We choose to employ a more detailed notation $\Gamma_{\Lambda_0 \to \Lambda}$ for this distribution. It has the same meaning as $\Gamma(\Lambda)$ in eq. (14) but explicitly contains the pre-kick rapidity $\Lambda_0$. The new notation highlights its physical meaning: $\Gamma_{\Lambda_0 \to \Lambda}$ is the density of the probability that the polaron with the initial rapidity $\Lambda_0$ will acquire the rapidity $\Lambda$ after the kick with the impulse $\Delta P$.

The explicit expression for the rapidity distribution is the first main result of the present paper. It reads

$$\Gamma_{\Lambda_0 \to \Lambda} = \frac{\Delta P^2}{(\Lambda - \Lambda_0)^2 \, Q'(\Lambda_0)} \mathrm{Re} \int\limits_0^\infty \frac{dx}{\pi} e^{-i\,\Delta P\,x} \det\left(1 + \hat{K}\right). \tag{17}$$

Here $\det\left(1 + \hat{K}\right)$ is a Fredholm determinant, and the operator $\hat{K}$ acts on $L^2[-p_F, p_F]$ and has an integrable kernel

$$K(q,q') = (\Lambda_0 - \Lambda)^2 \, \frac{e^{-ix(q+q')/2}}{\pi \sqrt{(\alpha q / p_F - \Lambda_0)^2 + 1} \, \sqrt{(\alpha q' / p_F - \Lambda_0)^2 + 1}} \, \frac{e(q) - e(q')}{q - q'} \, p_F, \tag{18}$$

$$e(q) = \frac{e^{ixq} - e^{ix\Lambda p_F / \alpha} e^{-x p_F / \alpha}}{\alpha q / p_F - \Lambda - i} - \frac{e^{ixq}}{\Lambda_0 - \Lambda}. \tag{19}$$

Note that $\hat{K}$ depends on $x$, $\Lambda_0$, $\Lambda$ and $\alpha$ as parameters. Note also a shorthand notation $Q'(\Lambda) = \partial Q / \partial \Lambda$ introduced in eq.(17).

One can view the Fredholm determinant as a thermodynamic limit of a finite determinant with matrix elements $\delta_{ij} + (2\pi/L)K(q_i, q_j)$, where $q_i, q_j \in [-p_F, p_F]$ are quantized momenta with momentum quantum $2\pi/L$. More details on properties of Fredholm determinants and an algorithm for their effective numerical computation can be found in ref. [66]. The derivation of eq. (17) is presented in Appendix B.

Since the polaron momentum $Q$ is a more intuitive quantity compared to the polaron rapidity $\Lambda$, we also introduce the polaron momentum distribution $\Gamma_{Q_0 \to Q}$. Its physical meaning is as follows: $\Gamma_{Q_0 \to Q}$ is the density of the probability that the polaron with the initial momentum $Q_0$ will acquire the momentum $Q$ after the kick with the impulse $\Delta P$. The two distributions are related as

$$\Gamma_{Q_0 \to Q} = \Gamma_{\Lambda_0 \to \Lambda} / Q'(\Lambda). \tag{20}$$

Plots of the momentum distribution for various impulses and initial polaron momenta is shown in Fig. 3. The distribution can feature two related effects, the Bragg reflection from the edge of the Brillouin zone and power-law singularities. Both of them are discussed in detail in the next subsection.

The polaron momentum can be hard to measure in the experiment since it is shared between the impurity and the accompanying disturbance of the Fermi sea. The polaron rapidity also does not have a direct operational meaning.[1] In contrast, the polaron velocity coincides with the velocity of the impurity and thus can be readily measured. One can straightforwardly obtain the velocity distribution $\Gamma_{v_0 \to v}$ from the rapidity distribution (17) and expression (11) for $v(\Lambda)$, keeping in mind that the latter map is two-to-one and thus folding should be employed.

To compute the average polaron velocity in the post-quench equilibrium state, one should convolve $v(\Lambda)$ with $\Gamma_{\Lambda_0 \to \Lambda}$ according to eq. (14). In practice, the integration in eq. (14) is performed over a deformed contour in the complex plane, analogously to those ref. [33]. The result is shown in Fig. 4.

In ref. [33] the average polaron velocity was calculated for a different quench protocol, see Fig. 4 for comparison. There the initial pre-quench state consisted of the undisturbed gas of noninteracting fermions and the impurity with the momentum $\Delta P$ that did not interact with the fermions. At $t = 0$ the interaction between the impurity and the fermions was turned on. Physically, this quench protocol described the injection of a bare impurity into the initially undisturbed Fermi sea, with the subsequent dressing of the impurity by excitations that eventually led to the formation of the polaron. In contrast, in the protocol under study the polaron exists from the outset, the quench leading in the change of the polaron momentum. The two protocols generally result in quite different values of the equilibrium polaron velocity, particularly for larger interactions strength, as can be seen in Fig. 4. At the same time, they coincide for large values of $\Delta P$ exceeding the uncertainty of the impurity momentum in the polaron state.

## 3.4 Two-sided power-law singularity of the distribution

The singularity in the distribution $\Gamma_{Q_0 \to Q}$ is present for the range of parameters depicted in the top right panel of Fig. 3 and analytically specified below in eq. (25). It can be obtained from

---

[1] Note that while the distribution of pseudomomenta $k_l$ in the post-quench state can be measured in the free-expansion experiments [67–70], such measurements can hardly give much information about the polaron since $k_l$ differ from the momenta of noninteracting fermions only by $O(1/N)$ terms, see eq. (5). This is because the impurity disturbs the otherwise noninteracting Fermi gas only locally. However, we note a related recent ref. [71] where a small patch of a one-dimensional gas has been instantly cut out, with an access to local properties within this patch. It remains to be seen whether the latter approach can be helpful to study polarons.

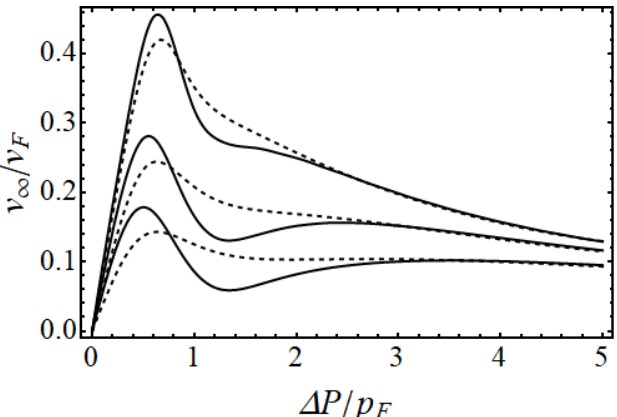

Figure 4: Steady state polaron velocity established after the kick as a function of the impulse (solid lines). The initial polaron momentum is $Q_0 = 0$. Different curves correspond to different coupling constants $2\pi/\alpha = 3, 6, 10$ (from top to bottom). For comparison, the results for a different quench protocol are shown (dashed lines) [33], where a bare impurity is injected in the initially undisturbed fluid with the momentum $\Delta P$.

the asymptotics of the Fredholm determinant, see Appendix C. The result reads

$$
\Gamma^{\text{sing}}_{Q_0 \to Q} =
\begin{cases}
\dfrac{C_+}{(Q - Q_{\text{sing}})^{1-(\xi_m^-)^2-(\xi_m^+)^2}}\,, & Q > Q_{\text{sing}}\,, \\[3ex]
\dfrac{C_-}{(Q_{\text{sing}} - Q)^{1-(\xi_m^-)^2-(\xi_m^+)^2}}\,, & Q < Q_{\text{sing}}\,.
\end{cases}
\tag{21}
$$

The position of the singularity $Q_{\text{sing}}$ is determined by the value of $(Q_0 + \Delta P)$, namely,

$$
Q_{\text{sing}} = Q_0 + \Delta P - 2 m p_{\text{F}}\,,
\tag{22}
$$

where the integer $m$ is chosen such that

$$
Q_{\text{sing}} \in [-p_{\text{F}}, p_{\text{F}}]\,.
\tag{23}
$$

The exponent of the singularity is constructed from

$$
\xi_m^\pm = \frac{1}{\pi}\Big(\arctan(\Lambda_0 \mp \alpha) - \arctan(\Lambda \mp \alpha)\Big) - m\,,
\tag{24}
$$

where $\Lambda = \Lambda(Q)$ and $\Lambda_0 = \Lambda(Q_0)$. The constants $C_\pm$ are given in Appendix C, see eq. (C.7). The true distribution $\Gamma_{Q_0 \to Q}$ is close to $\Gamma^{\text{sing}}_{Q_0 \to Q}$ for polaron momenta in the vicinity of $Q_{\text{sing}}$.

From the mathematical point of view, the singularity (21) is similar in origin to the threshold X-ray singularity [72]. While there is no any threshold here, the amplitudes $C_\pm$ of the left and right parts of the singularity differ, therefore it resembles two threshold singularities glued together. We adopt the term *two-sided* for such type of singularity.

The condition for the existence of the singularity is that the exponent in the asymptotics (21) is positive,

$$
1 - (\xi_m^-)^2 - (\xi_m^+)^2 > 0\,.
\tag{25}
$$

In general, this is a complicated nonlinear condition, see Fig. 3 for illustration. However, one simple fact about it can be easily obtained: the condition is never satisfied when $|m| \geq 2$. Therefore, in fact only $m = 0$ or $m = \pm 1$ are allowed. Let us discuss these two cases separately.

When $m = 0$, the kick acts within the main Brillouin zone, i.e. $Q_0 + \Delta P \in [-p_F, p_F]$. In this case the singularity corresponds to the process where the whole impulse of the kick is transferred to the polaron, with no Fermi sea excitations created. The smaller is the impulse $\Delta P$, the more probability weight is concentrated in the vicinity of the singularity, see Fig. 3. In the limit of vanishing $\Delta P$, the singularity turns into the delta-function embracing all the weight, i.e.

$$\Gamma_{Q_0 \to Q} \xrightarrow{\Delta P \to 0} \delta(Q - Q_0), \tag{26}$$

see Appendix C for the proof.

When $m = \pm 1$, the kick drives the system through the boundary between two Brillouin zones. In this case the Bragg reflection occurs: The impulse of the kick is shared between the polaron and the center-of-mass motion of the Fermi sea, the latter acquiring the momentum $\pm 2p_F$. Again, no Fermi sea excitations are created at $Q = Q_{\text{sing}}$. One can see from Fig. 3 that in this case the singularity carries a relatively low share of probability weight, with the majority weight being carried by a broad peak on the opposite side of the Brillouin zone.

## 4 Discussion and outlook

We have calculated the polaron momentum distribution established after a kick. The kick can be thought as a limiting case of a more general external driving with the force $F$ applied to the impurity for the time interval $\tau$, the acquired impulse being given by $\Delta P_\tau = F \tau$. In one dimension, such driving in general can lead to Bragg reflection from the boundary of the emergent Brillouin zone of the fluid, an effect first theoretically predicted for adiabatic driving [20, 60, 61] and subsequently observed for a finite driving force in an experiment with ultracold atoms [11]. We are able to rigorously determine the conditions for Bragg reflection in our setting. In particular, in contrast to the conventional Bragg reflection from crystals, here such reflection is operational only between neighbouring Brillouin zones, with the only available momentum change equal to $\pm 2p_F$.

In the adiabatic limit of fixed $\Delta P$ and $F = \Delta P / \tau \to 0$ the polaron was predicted to experience Bloch-like oscillations [20, 60, 61] with the polaron momentum $Q \simeq Ft \mod (2p_F)$ (see also [18, 22, 73–75]). Adiabatic driving can be emulated by a periodic sequence of small kicks. Whenever the interval between the kicks exceeds the polaron relaxation time, our result for the polaron momentum distribution can be applied after each kick. Since in the limit of the small impulse the distribution approaches the delta-function, see eq. (26), the above simple picture of Bloch-like oscillations is restored. This is consistent with the fact that Bloch-like oscillations are particularly robust for polarons that are heavier than the host particles [18, 22, 76, 77] (the effective mass of the polaron was calculated in [47]; it always exceeds the mass of the host fermion).

The two-sided power-law singularities that show up in the polaron momentum distribution correspond to processes where no host medium excitations are created, analogously to such genuinely solid state effects like X-ray singularities [72] or Mössbauer effect [78]. This highlights the peculiarity of one-dimensional fluids that originates from geometrically-enhanced quantum correlations.

An interesting question for further exploration is to what extent the qualitative picture established here survives the breakdown of the integrability. We note in this respect that the results obtained in integrable systems are often quite robust and do not change quantitatively away from the integrable point [22, 79].

From the experimental perspective, it is important to understand to what extent one can relax the idealized conditions adopted in the present study, most importantly – zero temperature and infinite relaxation time limits. Here we confine ourselves by brief remarks in this

respect. One expects that for intermediate coupling strengths, $\alpha \sim 1$, the equilibration typically occurs on time scales of a few Fermi times, as confirmed by numerical studies [16, 79]. Analogously, one expects that for temperature well below the Fermi energy all thermal effects are exponentially suppressed. More refined estimates can be obtained from a quantitative exploration of the polaron dynamics in the McGuire model at finite times and temperatures, which constitutes a promising direction for future research.

Finally, we note that it could be interesting to study the dynamics of the attractive polaron which is known to have a richer phenomenology compared to the repulsive case [52].

## A  Kick as a quantum quench

In order to rigorously define the kick in the translation-invariant system, one introduces a more general Hamiltonian

$$H_\Phi = \frac{1}{2}\left(P_{\text{imp}} - \Phi\right)^2 + \frac{1}{2}\sum_{j=1}^{N} P_j^2 + \frac{2p_F}{\alpha}\sum_{j=1}^{N}\delta(x_j - x_{\text{imp}}),\tag{A.1}$$

depending on the parameter $\Phi$. This Hamiltonian is solvable by Bethe Ansatz for arbitrary value of $\Phi$ (not necessarily an integer of the momentum quantum), with eigenstates $|\mathbf{n}, M\rangle_\Phi$ of the form (4) and the equation on the rapidity that can be obtained from eq. (8) by substituting $(2\pi/L)M = P_{\text{tot}} \to P_{\text{tot}} - \Phi$. If the value of $\Phi$ is an integer of momentum quanta $2\pi/L$, the eigenstates $|\mathbf{n}, M\rangle_\Phi$ are related to the eigenstates $|\mathbf{n}, M\rangle = |\mathbf{n}, M\rangle_{\Phi=0}$ in a simple way,

$$|\mathbf{n}, M\rangle_\Phi = e^{i\Phi x_{\text{imp}}}|\mathbf{n}, M\rangle,\tag{A.2}$$

which can be verified directly by applying $H_\Phi$ to both sides of the above relation.

The "kick" protocol of impurity preparation described in Section 3.2 corresponds to preparing the system in an eigenstate of the Hamiltonian $H_\Phi$ with $\Phi = \Delta P$ and subsequently quenching the value of $\Phi$ to zero. In view of the relation (A.2), such quantum quench results in the state (16).

## B  Momentum distribution: Derivation

### B.1  Preliminary considerations

To lighten the notations we employ the convention,

$$p_F = 1,\tag{B.1}$$

throughout the rest of the Appendix. One can always restore $p_F$ by dimensionality.

In this section we outline the derivation of the rapidity distribution defined as

$$\Gamma(\Lambda) = \sum_{|\mathbf{n},M\rangle}\left|\langle \mathbf{n}, M|e^{i\Delta P x_{\text{imp}}}|\mathbf{n}^0, M^0\rangle\right|^2 \delta(\Lambda - \Lambda_{\mathbf{n},M}),\tag{B.2}$$

cf. eqs. (15),(16). We use the technique employed earlier to calculate the Green's function of the impurity [51, 57, 80].

It turns convenient to introduce functions $k(n, \Lambda)$ and $\delta(n, \Lambda)$,

$$k(n, \Lambda) = \frac{2\pi}{L}\left(n - \frac{\delta(n, \Lambda)}{\pi}\right),\qquad \delta(n, \Lambda) = \frac{\pi}{2} - \arctan\left(\Lambda - \frac{2\pi}{L}\alpha n\right).\tag{B.3}$$

They can be used to specify solutions of Bethe equations. Indeed, $k(n_l, \Lambda_{\mathbf{n},M}) = k_l$ and $\delta(n_l, \Lambda_{\mathbf{n},M}) = \delta_l$ are, respectively, the $l$'th Bethe pseudomomentum and phase of the eigenstate $|\mathbf{n}, M\rangle$, *cf.* eqs. (5),(6).

Note that eq. (9) for $Q(\Lambda)$ is in fact obtained as the thermodynamic limit of

$$Q(\Lambda) = \frac{2\pi}{L} \sum_{n=-(N-1)/2}^{(N+1)/2} \left( \frac{1}{2} - \frac{1}{\pi} \delta(n, \Lambda) \right). \tag{B.4}$$

This equation along with eq. (B.3) imply the identity

$$\sum_{l=1}^{N+1} \frac{\partial k}{\partial \Lambda}(n_l, \Lambda) = Q'(\Lambda) + O(1/N), \qquad Q'(\Lambda) \equiv \frac{\partial Q(\Lambda)}{\partial \Lambda}, \tag{B.5}$$

valid for an arbitrary state $|\mathbf{n}, M\rangle$ from the bottom of the spectrum.

The importance of the quantity $\partial k/\partial \Lambda$ stems from the identity

$$\frac{\partial k}{\partial \Lambda}(n_l, \Lambda_{\mathbf{n},M}) = \frac{2}{L} (\sin \delta_l)^2 + O(1/N^2), \tag{B.6}$$

that follows from the complete set of Bethe equations. Thanks to this identity, $\partial k/\partial \Lambda$ enters matrix elements between eigenstates (4).

## B.2 Matrix element

The matrix element entering eq. (B.2) is obtained from the determinant representation (4) of the eigenfunction [50]. It reads

$$\left| \langle \mathbf{n}, M | e^{i\Delta P x_{\text{imp}}} | \mathbf{n}^0, M^0 \rangle \right|^2 = \delta_{M, M^0 + \Delta M} \left( \frac{\Delta P}{\Lambda_{\mathbf{n},M} - \Lambda_0} \right)^2 \frac{1}{Q'(\Lambda_0) Q'(\Lambda)} (\det D)^2$$

$$\times (\Lambda_{\mathbf{n},M} - \Lambda_0)^{2N+2} \prod_{l=1}^{N+1} \frac{\partial k}{\partial \Lambda}(n_l, \Lambda) \prod_{l=1}^{N+1} \frac{\partial k}{\partial \Lambda}(n_l^0, \Lambda_0). \tag{B.7}$$

Here $D$ is $(N+1) \times (N+1)$ Cauchy matrix constructed from the two sets of pseudomomenta corresponding to eigenstates $|\mathbf{n}^0, M^0\rangle$ and $|\mathbf{n}, M\rangle$,

$$D = \left\| \frac{1}{k(n_l, \Lambda_{\mathbf{n},M}) - k(n_{l'}^0, \Lambda_0)} \right\|_{l,l'=1,2,\dots,(N+1)}. \tag{B.8}$$

Due to momentum conservation the matrix element is nonzero for a single value of $M$ given by

$$M = M^0 + \Delta M, \qquad \Delta M \equiv \frac{L}{2\pi} \Delta P. \tag{B.9}$$

It should be reminded that $\Lambda_0 \equiv \Lambda_{\mathbf{n}^0, M^0}$.

## B.3 From sum over eigenstates to sum over independent integers

The next step is to replace the summation over eigenstates in eq. (B.2) by the summation over independent integers $n_l$, $l = 1, 2, \dots$. This is done as follows:

$$\sum_{|\mathbf{n},M\rangle} \longrightarrow \frac{1}{(N+1)!} \sum_{n_1} \sum_{n_2} \dots \sum_{n_{N+1}} \theta_M(\mathbf{n}). \tag{B.10}$$

Here each $n_l$ runs over all integers, $M$ is fixed according to eq. (B.9), the prefactor $1/(N+1)!$ accounts for permutations within the set $\mathbf{n}$, the function $\theta_M(\mathbf{n})$ equals 1 provided the constraint (3) is satisfied and 0 otherwise, and terms where at least two integers are equal vanish automatically thanks to the determinant $\det D$ in the matrix element (B.7).

## B.4 Handling $(\det D)^2$

The Cauchy-Binet theorem allows one to convert $(N+1)$ sums over $(\det D)^2$ to a single determinant,

$$
\frac{1}{(N+1)!} \sum_{n_1} \sum_{n_2} \ldots \sum_{n_{N+1}} \delta(\Lambda - \Lambda_{\mathbf{n},M}) (\det D)^2 \prod_{l=1}^{N+1} f(n_l, \Lambda_{\mathbf{n},M})
$$

$$
= \delta(\Lambda - \Lambda_{\mathbf{n},M}) \det \left\| \sum_{n=-\infty}^{\infty} \frac{f(n,\Lambda)}{\left(k(n,\Lambda) - k(n_l^0, \Lambda_0)\right)\left(k(n,\Lambda) - k(n_{l'}^0, \Lambda_0)\right)} \right\|_{l,l'=1,2,\ldots,(N+1)} , \quad \text{(B.11)}
$$

where $f(n,\Lambda)$ is an arbitrary function.

## B.5 Factorized representation of $\delta(\Lambda - \Lambda_{\mathbf{n},M})$

Since the Bethe equation (8) has a single solution $\Lambda = \Lambda_{\mathbf{n},M}$ provided $\mathbf{n}$ and $M$ satisfy the constraint (3), and no solutions otherwise, one can rewrite $\theta_M(\mathbf{n})\,\delta(\Lambda - \Lambda_{\mathbf{n},M})$ as follows:

$$
\theta_M(\mathbf{n})\,\delta(\Lambda - \Lambda_{\mathbf{n},M}) = \int_{-\infty}^{+\infty} \frac{dx}{2\pi} Q'(\Lambda)\, e^{ix\left(\sum_{l=1}^{N+1}\left(k(n_l,\Lambda) - k(n_l^0,\Lambda_0)\right) - \Delta P\right)} . \quad \text{(B.12)}
$$

Here the momentum conservation and the identity (B.5) has been employed. The exponential in the integrand can be factorized, which will prove useful in what follows.

## B.6 Combining the pieces

We combine eqs. (B.7), (B.5), (B.10), (B.11) and (B.12) to obtain

$$
\Gamma = \frac{\Delta P^2}{(\Lambda - \Lambda_0)^2\, Q'(\Lambda_0)} \operatorname{Re} \int_0^{\infty} \frac{dx}{\pi} e^{-i\,\Delta P\, x} \det \mathcal{A} , \quad \text{(B.13)}
$$

where the $(N+1) \times (N+1)$ matrix $\mathcal{A}$ has matrix elements

$$
\mathcal{A}_{ll'} = (\Lambda - \Lambda_0)^2 \sqrt{\frac{\partial k}{\partial \Lambda}(n_l^0, \Lambda_0)} \sqrt{\frac{\partial k}{\partial \Lambda}(n_{l'}^0, \Lambda_0)}\, e^{-ix\left(k(n_l^0,\Lambda_0) + k(n_{l'}^0,\Lambda_0)\right)/2}
$$

$$
\times \sum_{n=-\infty}^{\infty} \frac{\partial k}{\partial \Lambda}(n,\Lambda) \frac{e^{ix\,k(n,\Lambda)}}{\left(k(n,\Lambda) - k(n_l^0,\Lambda_0)\right)\left(k(n,\Lambda) - k(n_{l'}^0,\Lambda_0)\right)} . \quad \text{(B.14)}
$$

For $l \neq l'$ the second line of the above equation can be reorganized as

$$
\frac{e\left(k(n_l^0,\Lambda_0)\right) - e\left(k(n_{l'}^0,\Lambda_0)\right)}{k(n_l^0,\Lambda_0) - k(n_{l'}^0,\Lambda_0)} , \quad \text{(B.15)}
$$

with

$$
e\left(k(n_l^0,\Lambda_0)\right) = \sum_{n=-\infty}^{\infty} \frac{\partial k}{\partial \Lambda}(n,\Lambda) \frac{e^{ix\,k(n,\Lambda)}}{k(n,\Lambda) - k(n_l^0,\Lambda_0)} . \quad \text{(B.16)}
$$

This function has a nice thermodynamic limit that can be obtained by presenting the sum as a contour integral, a technique described in refs. [51, 80]. The result is given by eq. (19). For diagonal entries, $l = l'$, eq. (B.15) still can be used if interpreted in the l'Hôpital sense, with extra care required to keep both the O(1) and O(1/N) terms:

$$
\mathcal{A}_{ll} = 1 + (\Lambda - \Lambda_0)^2\, e^{-ixk} \frac{\partial k}{\partial \Lambda} \partial_k e(k)\Big|_{k=k(n_l^0,\Lambda_0)} + O(1/N^2) . \quad \text{(B.17)}
$$

Combining eqs. (B.13), (B.14), (B.15) and (19) one arrives at eq. (17).

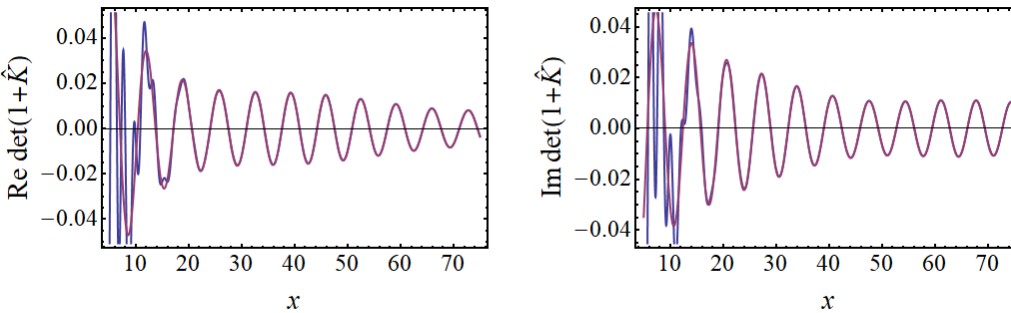

Figure 5: Comparison of the exact Fredholm determinant (18) (blue) and its asymptotic expression (C.1) (magenta) for $\Lambda_0 = 0$, $\Lambda = 10$, $\alpha = 3$.

# C  Asymptotic analysis of the singularity

## C.1  Asymptotics of the Fredholm determinant

The singularities in the rapidity and momentum distributions stem from the asymptotic behavior of $\det\left(1 + \hat{K}\right)$ for large $x$. The latter is given by

$$\det\left(1 + \hat{K}\right) \simeq \sum_{m=-1}^{1} \frac{C_{\Lambda_0,\Lambda}[\xi_m]}{(2i)^{(\xi_m^-)^2}(-2i)^{(\xi_m^+)^2}} \frac{e^{ix(Q-Q_0+2m)}}{x^{(\xi_m^-)^2+(\xi_m^+)^2}}, \quad x \gg 1/p_F, \tag{C.1}$$

where $\xi_m$ is a function given by

$$\xi_m(q) = \frac{1}{\pi}\Big(\arctan(\Lambda_0 - \alpha q) - \arctan(\Lambda - \alpha q)\Big) - m, \tag{C.2}$$

$\xi_m^\pm$ equals $\xi_m(\pm 1)$ (which is consistent with the definition (24) in the main text), the functional $C_{\Lambda_0,\Lambda}[\xi]$ reads

$$C[\xi] = \left(G(1 - \xi^-)\,G(1 + \xi^+)\right)^2 (2\pi)^{\xi^- - \xi^+} e^{-\tilde{C}_{\Lambda_0,\Lambda}[\xi]}, \tag{C.3}$$

with the subscript $m$ in $\xi_m(q)$, $\xi_m^\pm$ omitted for brevity, $G(a)$ is the Barnes function, and

$$\tilde{C}_{\Lambda_0,\Lambda}[\xi] = \int_{-1}^{1} dq\, \frac{\xi(q)\cot\left(\pi\xi(q)\right)}{1 + (\Lambda - \alpha q)^2} - \frac{2\alpha}{\Lambda_0 - \Lambda}\int_{-1}^{1} dq\,\xi(q) + \frac{1}{2}\int_{-1}^{1} dq \int_{-1}^{1} dq'\left(\frac{\xi(q') - \xi(q)}{q' - q}\right)^2$$
$$- \int_{-1}^{1} dq\, \frac{\xi(-1)^2 - \xi(q)^2}{1 + q} - \int_{-1}^{1} dq\, \frac{\xi(1)^2 - \xi(q)^2}{1 - q}. \tag{C.4}$$

The asymptotics (C.1) is derived with the help of the standard soft mode resummation technique [81–84] (see section 5 in ref. [80] for a pedagogical introduction).

It should be emphasized that we have restricted the summation over $m$ in eq. (C.1) by three terms $m = 0, \pm 1$. Only these terms can be relevant for the singularity of the distribution, as discussed in Section 3.4.

Importantly,

$$C_{\Lambda_0,\Lambda}[\xi] = C_{\Lambda,\Lambda_0}[-\xi], \tag{C.5}$$

thanks to the following property of the Barnes function:

$$G(1 - z) = \frac{G(1 + z)}{(2\pi)^z}\exp\left(\pi\int_0^z x\cot(\pi x)\,dx\right). \tag{C.6}$$

We compare the asymptotics (C.1) to the exact Fredholm determinant (18) in Fig. 5. One can see an excellent agreement for sufficiently large $x$. High frequency oscillations at small $x$ that are not captured by the asymptotics (C.1) stem from the term $e^{ix\Lambda p_F/\alpha}e^{-|x|p_F/\alpha}$ in eq. (19). They are visible only for large values of $\Lambda$.

## C.2   Singularity of the momentum distribution

Plugging the asymptotics (C.1) into eq.(17) and performing the integration over $x$, one obtains eq. (21) with

$$C_\pm = \frac{\Delta P^2}{(\Lambda - \Lambda_0)^2\, Q'(\Lambda_0)Q'(\Lambda)}\, \frac{\Gamma\big(1 - (\xi_m^-)^2 - (\xi_m^+)^2\big)}{\pi\, 2^{(\xi_m^-)^2 + (\xi_m^+)^2}}\, C_{\Lambda_0,\Lambda}[\xi_m]\, \sin\big(\pi\, (\xi_m^\mp)^2\big). \qquad (C.7)$$

Here $\Gamma(a)$ is the gamma-function and $C_{\Lambda_0,\Lambda}[\xi_m]$, $\xi_m$, $\xi_m^\pm$ are defined in the previous subsection, with $m$ chosen to satisfy the condition (23). The latter rule implies that a single term from the sum in eq. (C.1) contributes to the singularity. Note also that the argument of the gamma-function is always in the interval $(0,1)$ thanks to the condition (25).

The limit $\Delta P \to 0$ deserves a separate consideration. In this limit $m = 0$, the position of the singularity $Q_{\text{sing}}$ approaches $Q_0$, and $\xi_0(q), \xi_0^\pm \to 0$, $C_{\Lambda_0,\Lambda}[\xi_0] \to 1$. Therefore the Fourier transform of the asymptotics (C.1) leads to the delta-function,

$$\lim_{\Delta P \to 0} \Gamma^{\text{sing}}_{Q_0 \to Q} = \frac{1}{\pi}\text{Re}\frac{i}{(Q - Q_0 + i0)} = \delta(Q - Q_0), \qquad (C.8)$$

where the latter equality is the Plemelj-Sokhotski formula. The normalization condition implies that $\Gamma^{\text{sing}}_{Q_0 \to Q} = \Gamma_{Q_0 \to Q}$ in the limit $\Delta P \to 0$. This way one arrives at eq. (26).

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
