# Peer review of "One-dimensional Fermi polaron after a kick: two-sided singularity of the momentum distribution, Bragg reflection and other exact results"

_SciPost Physics, doi:SciPost Phys. 17, 063 (2024)_

## Round 1 · Referee Report · Anonymous (Referee 1) · 2024-5-14

Strengths

This paper studies, using an integrable model, the physics of an impurity coupled to a quantum bath (polaron) after a momentum kick has been applied to the impurity.

The question of the physics of an impurity in contact with a quantum bath is a question of considerable interest. In one dimension the vision of the combined features of the impurity and the bath as a single particle, which can be considered as propagating freely albeit with an effective mass somewhat breaks down. The resulting physics is much more complex and some of this has been already studied by various methods such as field theory, numerics and exactly solvable models. Experiments in cold atomic gases have provided a remarkable class of systems to investigate such physics.

I thus find the present paper, that addresses through an integrable model, the physics of an impurity subjected to a kick, both interesting and timely. The possibility to solve exactly is of course particularly precious in a such systems. The authors obtain from the exact solution a certain number of directly observable quantities such as the steady state velocity of the polaron.

The theoretical study is well done and the paper is well written, presenting the questions and the results in a clear way both in connection with previous theoretical studies but also making contact with potential observations.

Strenghts: - study of an interesting and experimentally relevant problem - an exact solution - computation of physically observable quantities - paper clearly written and positioning the problem well in connection with the previous literature on the subject.

Weaknesses

No specific weaknesses of the present paper, but some points that could be addressed by the authors and that would enhance the paper.

  • quite generally a slightly more detailed discussion of the physical consequences of the results would be useful.
  • given the recent measurements of rapidities of 1D quantum systems (see e.g. experiments by I. Bouchoule and D. Weiss) the authors could check/comment whether such measurements would bring interesting information in the case of their problem (or not) that measurements of the more macroscopic variables (such as the velocity of the impurity) would not give.
  • some more comments on Figure 4, in particular in connection with the two protocols given in the figure would be suitable.

Report

Given the above comments, I think that the paper provides interesting results on a timely and difficult problem, and is thus perfectly suitable for publication in Scipost. Since the paper is well written I do recommend publication in present form, but would suggest that the authors consider the above mentioned optional additions to the paper.

Recommendation

Publish (meets expectations and criteria for this Journal)

---

## Round 1 · Referee Report · Anonymous (Referee 2) · 2024-5-29

Strengths

1- Detailed, non-perturbative calculations of dynamical behaviour of quantum impurity in perfect 1D metal could be highly valuable benchmarks in a range of fields devoted to study this and many related problems.

2- Presentation is clear and quite rich on technical detail

Weaknesses

1- for experimental or numerical tests: does not yet exactly address a fully realistic regime, in that no guidance on temperatures at which theory could be validated is given, nor are the time scales estimated that would have to be reached at minimum in order to enter the stationary state of persistent impurity motion.

2- for theory validity: calculations are constrained to low-energy part of the many-body spectrum (what the authors term the "bottom of the spectrum"), but the exact cut-off used appears to be poorly specified. It would be good to correct this, and also to examine how the ultimate results might change, or not change, if the cut-off is varied. As unlikely as it may be on physical grounds, it is not inconceivable that the influence of highly excited states, albeit small, may fall off sufficiently slowly with rising cut-off that add the effects add up and impact the results - this needs to be checked.

Report

The present article is a very interesting potential addition to the research area of a the dynamics of mobile quantum impurity inside a many-body system. Its quasi-exact theory is non-perturbative and provides very valuable quantitatively testable predictions for numerical theory, and possibly even for experiments. If the specific concern highlighted are addressed adequately, this would measurably enrich the field.

Requested changes

1- Quantify, if possible, what temperature regime future experiments might have to reach in order to at least approach the theory presented in this paper.

2- Likewise, estimate, if possible, across which timescale the experiment, or many-body numerics for that matter, would need to track the impurity dynamics in order to approach the stationary regime, i.e. when would the off-diagonal elements that have been dropped in eq. (13) would have died off sufficiently?

3- Quantify to what extent the results shown are stable against variation of the cut-off in the many-body spectrum retained

Recommendation

Ask for minor revision

---

## Round 2 · Author Response

Warnings issued while processing user-supplied markup:

  • Inconsistency: Markdown and reStructuredText syntaxes are mixed. Markdown will be used.
    Add "#coerce:reST" or "#coerce:plain" as the first line of your text to force reStructuredText or no markup.
    You may also contact the helpdesk if the formatting is incorrect and you are unable to edit your text.

We thank the Referees for their positive reports. Below we address their specific comments.

Response to the first report.

  • quite generally a slightly more detailed discussion of the physical consequences of the results would be useful.

Some additional remarks on the experimental prospects are given in the footnote 1 on p. 9 and in the next-to-last paragraph of Sect. 4 on p. 12.

  • given the recent measurements of rapidities of 1D quantum systems (see e.g. experiments by I. Bouchoule and D. Weiss) the authors could check/comment whether such measurements would bring interesting information in the case of their problem (or not) that measurements of the more macroscopic variables (such as the velocity of the impurity) would not give.

We have briefly discussed this point in the footnote 1 on p.9. In our case, expansion experiments discussed and reported in refs. [67-70] could measure the distribution of pseudomomenta $k_l$. Note, however, that in the leading order of the thermodynamic limit this distribution is simply the Fermi-Dirac distribution. All the information about the polaron is contained in the O(1/N) corrections to the Fermi-Dirac distribution, and it is unlikely that this level of precision can be attained. On the other hand, the recent experiment [71] addresses the distribution locally. It is an interesting question whether this technique can be adapted to study polarons. One immediate complication would be the a priori unknown position of the polaron ( this might be countered by repeated heralded measurements). From a more fundamental point of view, the very concept of local distribution of pseudomomenta for polaron can require clarification and justification, since the variation of the Fermi gas density around the impurity is, in general, not slow compared to interparticle distance.

  • some more comments on Figure 4, in particular in connection with the two protocols given in the figure would be suitable.

A more detailed exposition of the injection protocol and its comparison to the kick protocol is given in the last paragraph of Section 3.3. on p.9

Response to the second report.

1- Quantify, if possible, what temperature regime future experiments might have to reach in order to at least approach the theory presented in this paper. 2- Likewise, estimate, if possible, across which timescale the experiment, or many-body numerics for that matter, would need to track the impurity dynamics in order to approach the stationary regime, i.e. when would the off-diagonal elements that have been dropped in eq. (13) would have died off sufficiently?

Roughly, the temperature must be well below the Fermi energy, while the time scale must exceed a few Fermi times, as we discuss in the next-to-last paragraph of Sect. 4 on p. 12. We plan to accurately answer these questions in a sequel to the present paper where the real-time dynamics at a finite temperature will be addressed.

3- Quantify to what extent the results shown are stable against variation of the cut-off in the many-body spectrum retained.

Firstly, let us stress that the above cut-off is not introduced as an explicit quantity and thus can not be varied in our calculations. Rather, it is introduced implicitly as follows: all eigenstates entering the sums in eqs. (13), (15) are treated as if they were from the bottom of the spectrum and thus satisfied eqs. (10), (11), (31) etc. The rationale behind such treatment is the assumption that higher-lying states with a thermodynamically large number of particle-hole excitations give a negligible contribution to these sums and thus are unimportant anyway. As the Referee correctly notes in the report, this assumption seems plausible on physical grounds but is not rigorously proven: while any individual term with O(N) particle-hole excitations is suppressed exponentially in N, as can be inferred from eq. (35), there are exponentially many such terms that, in principle, could add up to a finite contribution.

Apart from the physical intuition, the above assumption is supported by numerical tests of ref. [33,64], as we point out in the revised version (see a sentence below eq. (13)). The final resolution of this issue will be presented in the above-mentioned sequel to the present paper, where we will abandon the above assumption and rigorously account for finite-entropy states.

---

## Round 2 · List of Changes

1. A sentence added below eq. (13). It clarifies the grounds for considering only low-lying states.

  2. Footnote 1 added.

  3. A paragraph on the injection protocol added in the end of Sect. 3.3.

  4. A paragraph discussing rough estimates of relevant temperature and time scales added in the end of Sect. 4.

  5. Minor style and grammar improvements introduced.

  6. Refs. [64,67-71] introduced.

---

## Editorial Decision

published